# Clinical Efficacy of Catheter Ablation in the Treatment of Vasovagal Syncope

**DOI:** 10.3390/jcm11185371

**Published:** 2022-09-13

**Authors:** Lingping Xu, Yixin Zhao, Yichao Duan, Rui Wang, Junlong Hou, Jing Wang, Bin Chen, Ye Yang, Xianjun Xue, Yongyong Zhao, Bo Zhang, Chaofeng Sun, Fengwei Guo

**Affiliations:** 1Department of Cardiovascular Medicine, The First Affiliated Hospital of Xi’an Jiaotong University, Xi’an 710061, China; 2Department of Cardiovascular Medicine, Syncope Center, Xianyang Central Hospital, Xianyang 712000, China; 3Department of Neurology, The First Affiliated Hospital of Xi’an Jiaotong University, Xi’an 710061, China; 4Department of Cardiovascular Surgery, The First Affiliated Hospital of Xi’an Jiaotong University, Xi’an 710061, China

**Keywords:** syncope, vasovagal syncope, catheter ablation, ganglionated plexus, high-frequency stimulation

## Abstract

Catheter ablation of ganglionated plexi (GPs) performed as cardioneuroablation in the left atrium (LA) has been reported previously as a treatment for vasovagal syncope (VVS). However, the efficacy and safety of catheter ablation in the treatment of VVS remains unclear. The objective of this study is to explore the efficacy and safety of catheter ablation in the treatment of VVS and to compare the different ganglion-mapping methods for prognostic effects. A total of 108 patients with refractory VVS who underwent catheter ablation were retrospectively enrolled. Patients preferred to use high-frequency stimulation (HFS) (*n* = 66), and anatomic landmark (*n* = 42) targeting is used when HFS failed to induce a positive reaction. The efficacy of the treatment is evaluated by comparing the location and probability of the intraoperative vagal reflex, the remission rate of postoperative syncope symptoms, and the rate of negative head-up tilt (HUT) results. Adverse events are analyzed, and safety is evaluated. After follow-up for 8 (5, 15) months, both HFS mapping and anatomical ablation can effectively improve the syncope symptoms in VVS patients, and 83.7% of patients no longer experienced syncope (<0.001). Both approaches to catheter ablation in the treatment of VVS effectively inhibit the recurrence of VVS; they are safe and effective. Therefore, catheter ablation can be used as a treatment option for patients with symptomatic VVS.

## 1. Introduction

Vasovagal syncope (VVS), a neuro-mediated syndrome, is a disorder of cardiovascular autonomic regulation and is the most common cause of sudden and transient loss of consciousness caused by cerebral hypoperfusion [1]. Although syncope itself has a good prognosis, VVS causes physical injuries and poor quality of life [2]. The difference in heart rate and blood pressure with a positive HUT indicates that VVS can be divided into cardioinhibitory, vasodepressive, and mixed according to the pattern of hemodynamic changes during syncope attacks [3]. Treatment for VVS has been challenging. Conventional interventions (education, avoiding precipitating factors, and maintaining fluid and salt intake), orthostatic training, pharmacological treatments, and implantable rhythm devices have failed to show good clinical outcomes (25–65% recurrent syncope) [4,5,6]. The pathophysiological mechanisms that underlie VVS are complex and far from being fully elucidated. Pathological increases in vagal tone may play a significant role in VVS [2]. Posterior vagus ganglion neurons of the heart are mainly distributed in the epicardial fat pad, which transmits information between preganglionic and postganglionic nerve fibers and regulates cardiac rhythm and conduction [7,8]. The search for a more effective therapy aimed at achieving long-term suppression of vagal reflexes is therefore justified. Preliminary studies have shown that catheter ablation of the LA endocardial parasympathetic ganglion is effective in the treatment of refractory VVS and can achieve a good long-term prognosis [9,10,11]. Here, we describe 108 patients with VVS who underwent LAGP catheter ablation; in this study, we compared high-frequency stimulation (HFS) with anatomic markers and analyzed ablation safety and efficacy.

## 2. Materials and Methods

### 2.1. Patients

In total, 1425 patients with VVS were diagnosed by the syncope center from January 2018 to January 2021 in Xianyang Central Hospital, China, including 125 patients with VVS who received catheter ablation, in which 8 patients who were lost to follow-up and 9 patients with missing postoperative follow-up data. Finally, a total of 108 patients in our syncope center were retrospectively enrolled.

All patients had (1) fulfilled the diagnostic criteria of VVS proposed by the 2015 Heart Rhythm Society Guideline and underwent cardioneuroablation [4]; (2) there was no improvement in symptoms after drug treatment through optimal salt and liquid intake and physical back pressure training; (3) positive response to HUT; (4) the patients or their family members agreed to accept the study protocol and signed their informed consent.

Exclusion criteria were (1) structural cardiac or cardiopulmonary diseases (cardiac valvular diseases, severe aortic stenosis, acute myocardial infarction within 6 months, pulmonary embolism, pulmonary hypertension, and hypertrophic obstructive cardiomyopathy); (2) cerebrovascular diseases (vascular steal syndromes and seizures); (3) medication-related syncope (vasodilators, antipsychotics, and antidiabetics); (4) disease that affected autonomic nerve (diabetes mellitus and nervous system–related diseases) and (5) terminal diseases or New York Heart Association Class III or IV heart failure.

The protocols involving human participants were reviewed and approved by the Institutional Ethics Committee for Biomedical Research of Xianyang Central Hospital. The patients and participants provided written informed consent to participate in this study.

### 2.2. HUT

HUT was performed in the morning in patients in a fasting state. A SHUT-100 tilt table (SHUT-100, Standard Healthcare, Jiangsu Province, China) with a footboard for weight bearing and restraining belts was used for the procedure. Subjects underwent a passive phase (tilted at 70° for 30 min, if under 18 or over 75 years of age tilted at 60°). If no symptoms occurred during the passive phase, participants further underwent an additional provocative phase (0.5 mg sublingually administered nitroglycerin and continued to be tilted for an additional 20 min). Continuous ECG monitoring and noninvasive blood pressure measurements were performed. If after diagnosis by 2 or more experienced clinicians, the physician believed that the clinical presentation was consistent with VVS and the HUT nitroglycerin stimulation test result was negative, isoproterenol stimulation of HUT was added (starting with 1 μg/min and increasing by 1 μg/min every 5 min until reaching 3 μg/min, and ending when the average heart rate exceeded 20–25% of the baseline level; the maximum heart rate was not allowed to exceed 150 bpm). A positive response of HUT was defined when syncope or presyncope occurred in the presence of bradycardia (heart rate < 40 bpm) or abrupt hypotension (systolic blood pressure < 70 mmHg or diastolic blood pressure < 40 mmHg) as well as reproduction of the patient’s relevant clinical symptoms [3].

### 2.3. Preablation Preparation

Before the procedure, all medications were discontinued for at least 5 half-lives. All procedures were performed under local anesthesia. Pulse oximetry and blood pressure were monitored during the procedure. Routinely, the subclavian vein and femoral vein were punctured to place electrodes in the coronary sinus (CS) and right ventricle. Then we used the annular pulmonary vein catheter puncture into the LA through the atrial septum. The 3-dimensional geometry of the LA was conducted under the guidance of the Ensite-Navx Velocity 5.0 system (Abbott) or Carto3 Version 6.0 system (Johnson & Johnson). Target mapping and ablation were performed in the LA using a cold saline irrigated-tip ablation catheter. Patients were preferred to use HFS, and anatomic landmark targeting was used when HFS failed to induce a positive reaction. Patients were divided into an HFS group and an anatomic ablation group according to different operation modes. In the HFS group, HFS mapping was performed on the endocardial surface of the LA to search for positive response points and radiofrequency ablation. In the anatomical ablation group, the ablation endpoint was the disappearance of the vagal response in all GP anatomical regions (i.e., repeated stimulation or ablation in GP no longer showed a positive response).

During the operation, ventricular pacing was automatically output by a temporary pacemaker to avoid ventricular electrode connection with an EP4 stimulator or HFS by ventricular electrode output accidentally caused by the operator. To ensure cerebral perfusion, ablation was stopped when the blood pressure was lower than 70/40 mmHg as assessed by invasive blood pressure monitoring of the radial artery.

### 2.4. High-Frequency Stimulation-Guided Endocardial Catheter Ablation of GPs in the LA

HFS (50 Hz, 10–20 mA, 5 s) was applied by an electrophysiological stimulator (St Jude Medical EP4 electrophysiological stimulator) to identify five common GP distribution areas in the LA. GPs site could be anatomically located as follows: left superior GP (located in the superolateral area around the root of the left superior pulmonary vein, LSGP); left inferior GP (located in the inferoposterior area around the root of the left inferior pulmonary vein, LIGP); right anterior GP (located in the superoanterior area around the root of the right superior pulmonary vein, RAGP); right inferior GP (located in the inferoposterior area around the root of the right inferior pulmonary vein, RIGP) and coronary sinus electrodes from 3.4 to 5.6 corresponding to the CSMGP of the endocardial region of the LA [12]. In some patients, GP was widely distributed or displaced, and LIGP was extended to the anterior crest of the left upper pulmonary vein and the anterior wall between the left two lungs (Figure 1). RAGP extends to the top of the right superior pulmonary vein vestibule. The middle of the CS covers the coronal sinus electrode 1.2–7.8 corresponding to the endocardium (Figure 2). When GPs were stimulated by high frequency, an RR interval prolonged by more than 50% and a radial artery invasive pressure decreased by more than 5 mmHg were defined as positive reaction points (Figure 3). These positive reaction points were then ablated (power 35 W, temperature limit 43 °C, target force-time integral 400 gs). Each positive point was ablated for 30 s, which was repeated until reached the endpoint. HFS often induces atrial fibrillation. A temporary pacemaker was used if cardiac arrest occurred during the process (heart rate below 40 bpm). After marking positive reaction points, discharge ablation was performed one by one until the endpoint was reached (Figure 4).

### 2.5. Anatomically Guided Endocardial Catheter Ablation of GP in LA

The above GP distribution areas were located by ablation, and the 5 GP anatomical positions of the LA chamber were tested and ablated successively from the upper left to the left lateral left, lower left, right front, and lower right (power 35 W, temperature limit 43 ℃, target force-time integral 400 gs). Each positive point was ablated for 30 s, which was repeated until all five GPs reached the endpoint.

### 2.6. Postablation Follow-Up

Rivaroxaban 20 mg was taken daily after surgery. Previous medications, including beta blockers, fludrocortisones, and midodrine, were discontinued after the procedure. At a follow-up of 8 (5, 15) months after surgery, an outpatient review of ECG, cardiac ultrasound, 24 h ambulatory ECG monitoring, and HUT was performed. Syncope symptoms were assessed, and recurrent syncope and any associated physical impairment were recorded. Both recurrent syncope and any related physical injuries were carefully documented. Prodromes including transient dizziness, diaphoresis, or fatigue without loss of consciousness were not considered recurrent episodes of syncope. During the follow-up period, a change in HUT from positive (preoperatively) to negative, the lack of recurrence of syncope, or the presence of syncope with 50% fewer episodes than before was considered effective. Any associated perioperative adverse events were recorded in detail to assess safety.

### 2.7. Statistical Analysis

All data are reported as the mean ± SD for continuous variables and as the number of subjects (%) for categorical variables. Measurement data of skewness distribution are represented by Median (Q1, Q3). Independent sample t-tests were used to analyze differences between the two groups. Paired T-tests or Wilcoxon rank sum tests were used for comparisons of pre- and post-ablation measures. Categorical variables were compared using Pearson χ2 analysis. A two-sided *p* value < 0.05 indicated statistical significance. Data were analyzed using the SPSS statistical package for Windows, version 22.0 (IBM Corp., Armonk, NY, USA).

## 3. Results

### 3.1. Patient Characteristics

Comparison of baseline data between the two groups (Table 1). In total, 108 patients participated in this retrospective clinical study. There were no significant differences between the anatomic ablation group (*n* = 42) and the HFS group (*n* = 66). All patients successfully completed catheter ablation. The mean age of the patients was 51.2 ± 15.3 years, and 44.4% of them were males. The 108 patients were classified into the mixed depression type (62.0%), vascular depression type (38.0%), and cardiac depression type (0%) according to the preoperative HUT. Concurrent arrhythmias were as follows: atrial arrhythmias (paroxysmal atrial fibrillation, sustained atrial fibrillation, atrial premature beats, atrial tachycardia, atrial flutter): 41.7%; sinus bradycardia: 5.6%; intermittent atrioventricular block: 1.9%; ventricular arrhythmias (ventricular premature beat, ventricular tachycardia): 25.0%; supraventricular tachycardia (atrioventricular nodule reentrant tachycardia, atrial ventricular reentrant tachycardia): 17.6%.

### 3.2. Catheter Ablation

Detailed information about the procedures for the following sites is listed in Table 2: LSGP (71.3%), LIGP (17.6%), RAGP (53.7%), RIGP (19.4%), CSMGP (20.4%), no positive reaction (8.3%). The immediate heart rate increased by 15 (3, 30) bpm before and after intraoperative ablation. The recovery time of the sinoatrial node (SNRT) was shortened by 264 (140,359) ms, and the SNRT shortening rate was 19.4%. The invasive pressure of the radial artery decreased by 31.8 ± 22.7 mmHg during ablation, which indicates that there is a positive reaction to ablation.

### 3.3. Clinical Outcomes

At a follow-up of 8 (5, 15) months, the patients’ symptoms of syncope improved significantly after catheter ablation; 83.7% of patients had no recurrence of syncope (<0.001), and 81.5% had HUT results that turned negative Subgroup analysis showed that anatomical ablation and HFS ablation could effectively improve syncope symptoms (Table 3). During the 8 (5, 15) month follow-up, eight patients (11.9%) underwent a conversion from mixed suppression to vascular suppression (absence of heart rate inhibition and residual blood pressure inhibition). The number of syncope symptoms decreased in 76.9% of the patients compared with the previous attacks (Table 4). At 1-year follow-up, the number of syncope symptoms decreased in 84.3% of the patients compared with the previous attacks (Table 5). Both HFS mapping and anatomical ablation can effectively improve the syncope symptoms in VVS patients (Table 4). A Holter monitoring comparison of 108 VVS patients before and after catheter ablation showed that the mean heart rate (SDNN) of the slowest heart rate was statistically different (Table 6).

Two patients in the HFS group developed postoperative bradycardia with a ventricular rate of 45–55 bpm and hypotension of 80/50 mmHg. Dopamine and atropine were used to improve the blood pressure and heart rate, and sinus rhythm was restored 72 h later, with blood pressures above 90/60 mmHg. During the follow-up period, syncope or with recurrent precursory symptoms of syncope, such as dizziness, palpitation, and sweating, disappeared, and the HUT turned negative. There were no other surgery-related complications, including malignant arrhythmias, cardiac tamponade, pericarditis, symptoms associated with delayed gastric emptying, or death.

## 4. Discussion

The results revealed the following findings: (1) Symptoms associated with syncope were improved by both HFS and anatomic ablation-guided LA ganglion ablation in patients. A total of 83.7% of patients did not experience a recurrence of syncope or any related physical injury during a mean follow-up period of 8 (5, 15) months. (2) Catheter ablation has shown some effectiveness in the treatment of mixed inhibitory VVS, which can effectively relieve heart rate inhibition. (3) Many patients with VVS also have atrial fibrillation, idiopathic premature ventricular beats, and supraventricular tachycardia, and these patients should pay attention to preoperative screening of VVS. In this study, the sample size was significantly increased. Moreover, preoperative and postoperative HUT changes were added to the evaluation of efficacy. Patients with arrhythmias had no hemodynamic disorders and no malignant arrhythmias at the time of the arrhythmia attack. Moreover, no malignant arrhythmias were detected during the syncope episode. HUT induces a decrease in heart rate and/or blood pressure consistent with the patient’s clinical symptoms. Therefore, the patient’s symptoms were considered to be caused by VVS. Although two patients in our study developed arrhythmias after the invasive procedure, both recovered after drug treatment; consequently, catheter ablation of LAGP did not increase the incidence of sympathetic-related malignant arrhythmias. This unreproduced and non-controlled study is our preliminary experience demonstrating the efficacy and safety of LAGP ablation for the prevention of VVS.

The autonomic nervous system affects the function of the cardiovascular system by regulating the delicate balance between sympathetic and parasympathetic tension. VVS is caused by the imbalance of sympathetic and parasympathetic tension and the negative effect of pathologically increased vagal tone on cardiac conduction and vascular tension [2]. Therefore, some scholars have proposed achieving permanent endocardial denervation by catheter ablation of the epicardial ganglion of the atrial wall from the endocardial surface to treat patients with VVS [13,14]. Professor Pachon published articles in 2005 and 2011 demonstrating that catheter ablation can significantly improve the symptoms of patients with VVS [10,11]. The parasympathetic nerves were found to be more densely distributed in the atrial than the sympathetic nerves, with a proportion between 1.3:1 and 1.6:1, mainly located in the subendocardium of the myocardium [15]. Second, anatomical studies of the internal cardiac nervous system show that GP in the LA is mainly located around the root of the pulmonary veins [16]. Subsequently, Sun Wei reported that LAGP ablation showed good long-term clinical outcomes in a protocol that was performed only on the LA and not the right atrium or atrial septum [9].

In our study, the target and end points of catheter ablation were clear and feasible. A total of 91.7% of patients achieved a clear ablation endpoint. Patients followed up after ablation showed significant improvement in syncope or syncope-related symptoms such as dizziness and chest tightness, which severely affected daily life. The HUT review showed that 81.5% of patients changed from positive to negative; 79.1%, from mixed inhibition to negative; 11.9%, from mixed inhibition to vascular inhibition, suggesting that catheter ablation is more effective in the treatment of mixed inhibition of VVS and can effectively relieve the inhibition of heart rate. Some patients with VVS do not experience a syncope attack but have symptoms such as chest tightness, dizziness, blaumosis, palpitation, fatigue, sweating, etc., or have a simultaneous arrhythmia, which seriously affects daily life. Combined with the inducement, clinical manifestation, and positive HUT result, VVS can be diagnosed. For arrhythmias requiring catheter ablation, VVS should be routinely examined before surgery. For patients with obvious symptoms, modified LAGP ablation can be performed simultaneously [17]. This study showed that many patients with VVS also had atrial fibrillation, idiopathic premature ventricular beats, and supraventricular tachycardia. Therefore, attention should be given to the preoperative screening of VVS in these patients. In this study, 11.1% of patients had had a definite diagnosis of VVS without complications and received catheter ablation.

After the modified ablation of LAGP, sympathetic nerve tension increased, and there was a possibility of increased malignant arrhythmias. However, in this study, only part of the LA vagus nerve received the intervention, namely, modified ablation. The results of dynamic ECG showed that SDNN, the heart rate variability index, was shortened. After ablation of left atrial vagal ganglion, vagal tone decreased, and heart rate may increase. There was no difference in the maximum heart rate before and after the operation, and no malignant arrhythmia was observed after the operation. It is suggested that catheter ablation of LAGP does not increase the incidence of sympathetic-related malignant arrhythmias.

The comparison between the anatomical ablation group and the HFS group showed that there was no difference between the two groups in the intraoperative mapping of the LSGP, RAGP, LIGP, and RIGP, and the high positive rate of HFS mapping in the CSMGP [9]. The reason may be that the CSMGP is related to the location of the Marshall vein opening, and the location varies greatly. In the anatomical ablation group, CS_3, 4–5, 6_ and six electrodes were ablated and mapped, and the positions were relatively fixed. The HFS group had an advantage because repeated HFSn mapping was performed in a wide area involving CS_1,2–9,0_. In both groups, the positive rates of the LSGP and RAGP were significantly higher than those of the other GPs. Due to individual differences in GP distribution, the intraoperative mapping of these two GPs should be carefully expanded. Our center summarized the experience of catheter ablation in the treatment of VVS and suggested routine intervention of the LSGP and RAGP regardless of the positive results of the map. In addition, after the intervention of the RAGP, the incidence of vagal reflex in other parts of the ablation region decreased or ceased, which affected the clarity of the ablation endpoint. Therefore, ablation of the RAGP should be performed last to ensure that each GP can reach the definite endpoint.

There was no significant difference in most parameters between the anatomic ablation group and the HFS group, and both methods were optional. The advantages of HFS are that the mapping range is wide, the damage is small, the repeatability is high, and the mapping is accurate, but the stimulation intensity is weaker than the rf energy.

## 5. Conclusions

Catheter ablation may be a safe and effective in the treatment of VVS. Both HFS mapping and anatomical ablation can effectively improve the symptoms of patients with VVS, and LAGP ablation may be more effective in improving the components of cardiac inhibition.

## 6. Study Limitations

This study was a retrospective single-center sample. Although the sample size was large, it still had some limitations, and further multicenter, prospective, large-sample randomized controlled studies should provide evidence-based information for catheter ablation in the treatment of VVS. In addition, there is a selection bias as follows: patients who failed with HFS underwent the anatomical approach. Long-term follow-up is needed to determine whether nerve regeneration leads to the recurrence of syncope symptoms.

## Figures and Tables

**Figure 1 jcm-11-05371-f001:**
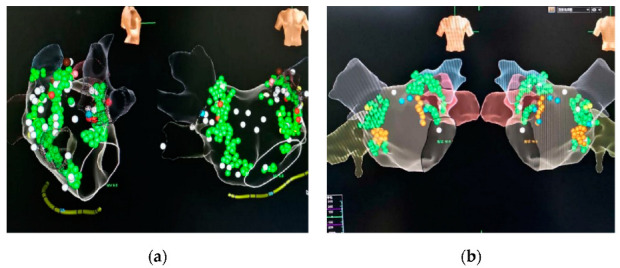
Distribution of GP. (**a**) High-frequency stimulation group. The green dots represent positive spots. The blood pressure and heart rate decreased significantly during high-frequency stimulation, which was widely distributed in the pulmonary vein vestibule. The white dots are negative spots; the heart rate and blood pressure did not change significantly during stimulation. The red dots represent cardiac arrest for >4 s at the time of stimulation. (**b**) Anatomical group. The green points are positive reaction points for anterior wall GP ablation, the yellow points are positive reaction points for posterior wall GP ablation, and the white and blue points are negative points.

**Figure 2 jcm-11-05371-f002:**
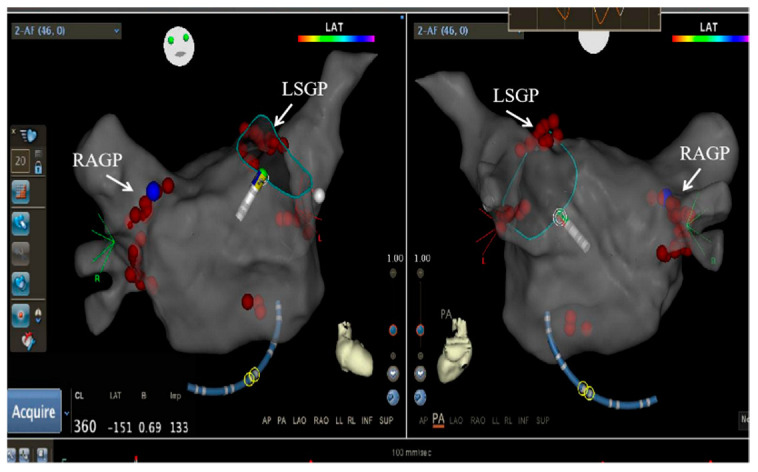
Distribution of ablation sites for anatomic ablation: LSG, RAGP, LIGP, RIGP, and CSMGP.

**Figure 3 jcm-11-05371-f003:**
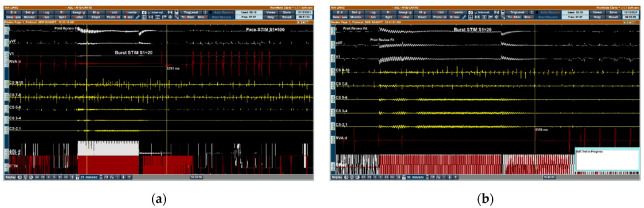
Ventricular arrest induced by high-frequency stimulation. (**a**) Upper left GPHFS induced 5251 ms of asystole. (**b**) Right anterior GPHFS induced 4350 ms of asystole.

**Figure 4 jcm-11-05371-f004:**
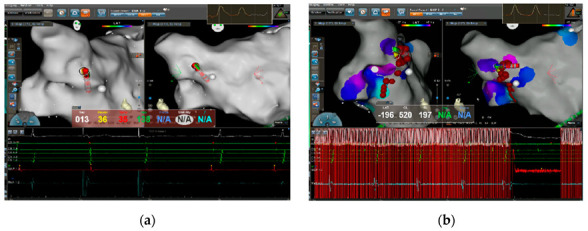
(**a**) During RAGP ablation, the heart rate decreased, the temporary pacemaker started pacing below the set frequency, and ablation continued until the positive reaction disappeared. (**b**) Repeated high-frequency stimulation verification after ablation showed no heart rate decline, reaching the ablation endpoint.

**Table 1 jcm-11-05371-t001:** Baseline characteristics of the study patients between the two groups.

	Anatomical Ablation Group (*n* = 42)	High-Frequency Stimulation Group (*n* = 66)	*p* Value
Age, years	47.9 ± 13.8	53.4 ± 15.8	0.066
Sex, female (%)	15 (35.7)	33 (50.0)	0.145
Diabetes, *n*	16	26	0.893
Serum creatinine, µmol/L	71.7 (61.3, 80.2)	70.0 (61.0, 81.9)	0.944
Left atrial diameter, mm	35.0 (31.0, 43.0)	38.0 (33.0, 46.0)	0.605
Left ventricular end diastolic diameter, mm	43.0 ± 7.3	45.0 ± 8.8	0.742
Left ventricular ejection fraction, %	63.7 ± 7.0	62.8 ± 7.6	0.749
Syncope burden			
Number of syncopal episodes in the preceding year	2.7 ± 1.9	2.5 ± 1.7	0.437
Number of precursory symptoms of syncope	23 (54.7)	36 (54.5)	0.982
Number of symptoms of syncope, *n* (%)	19 (45.2)	30 (45.5)	0.982
Complications			
Atrial arrhythmia*, *n* (%)	17 (40.5)	28 (42.4)	0.841
Sinus bradycardia, *n* (%)	2 (4.8)	4 (6.1)	1
Intermittent atrioventricular block, *n* (%)	1 (2.4)	1 (1.5)	1
Ventricular arrhythmias*, *n* (%)	12 (28.6)	15 (22.7)	0.494
Supraventricular tachycardia (AVRT, AVNRT), *n* (%)	7 (16.7)	12 (18.2)	0.84
Coronary atherosclerotic heart disease, *n* (%)	7 (16.7)	9 (13.6)	0.666
Hypertension, *n* (%)	6 (14.3)	15 (22.7)	0.28
Congenital heart disease, *n* (%)	1 (2.4)	5 (7.6)	0.401
Coronary artery spasm, *n* (%)	2 (4.8)	1 (1.5)	0.559
VVS types			
Mixed type	27 (64.3)	40 (60.6)	0.701
Vasodepressor type	15 (35.7)	26 (39.4)	0.701
Cardioinhibitory type	0	0	-

Data are expressed as the mean (SD), number (%), or *n*. Atrial arrhythmia*: paroxysmal atrial fibrillation, persistent atrial fibrillation, atrial premature beat, atrial tachycardia, atrial flutter. Ventricular arrhythmias*: Premature ventricular beat, ventricular tachycardia.

**Table 2 jcm-11-05371-t002:** Comparison of procedure parameter changes between the two groups.

	Anatomical Ablation Group *n* = 42	High-Frequency Stimulation Group *n* = 66	*p* Value
LSGP, *n* (%)	29 (69.0)	48 (72.7)	0.68
LIGP, *n* (%)	4 (9.5)	15 (22.7)	0.119
RAGP, *n* (%)	20 (47.6)	38 (57.6)	0.312
RIGP, *n* (%)	5 (11.9)	16 (24.2)	0.114
CSMGP, *n* (%)	4 (9.5)	18 (27.3)	0.029
Negative vagal response, *n* (%)	9 (21.4)	0 (0)	-
The ablation endpoint was defined and reached, *n* (%)	33 (78.6)	66 (100.0)	-

**Table 3 jcm-11-05371-t003:** Clinical data of the patients pre- and postablation.

		Preoperative *n* = 108	Postoperative *n* = 108	*p* Value
Total (*n* = 108)				
	HUT, *n* (%)			
	Mixed type	67	6	<0.001
	Vasodepressor type	41	14	<0.001
	Negative type	0	88	<0.001
	Syncope, *n* (%)	49	8	<0.001
HFS-Guided Ablation (*n* = 42)				
	HUT, *n* (%)			
	Mixed type	40	2	<0.001
	Vasodepressor type	26	14	0.023
	Negative type	0	50	<0.001
	Syncope, *n* (%)	30	0	<0.001
Anatomically Guided Ablation (*n* = 66)				
	HUT, *n* (%)			
	Mixed type	27	4	<0.001
	Vasodepressor type	15	0	<0.001
	Negative type	0	38	<0.001
	Syncope, *n* (%)	19	8	0.01

**Table 4 jcm-11-05371-t004:** Short-term follow-up results of syncope symptoms in the two groups.

	Anatomical Ablation Group	High-Frequency Stimulation Group	*p* Value
No recurrence of syncope, *n* (%)	11(26.2)	30(45.5)	0.002
Reduced syncope attacks, *n* (%)	8(19.0)	0(0)	
Improvement of precursory symptoms of syncope, *n* (%)	13(31.0)	21(31.8)	
No improvement, *n* (%)	10(23.8)	15(22.7)	

**Table 5 jcm-11-05371-t005:** One-year follow-up results of syncope symptoms in the two groups.

	Anatomical Ablation Group	High-Frequency Stimulation Group	*p* Value
No recurrence of syncope, *n* (%)	10(23.8)	28(42.4)	0.007
Reduced syncope attacks, *n* (%)	9(21.4)	2(3.0)	
Improvement of precursory symptoms of syncope, *n* (%)	18(42.9)	24(36.4)	
No improvement, *n* (%)	5(11.9)	12(18.2)	

**Table 6 jcm-11-05371-t006:** Changes in HR and HR variation after ablation during follow-up.

	Preoperative *n* = 108	Postoperative 8 (5, 15) Months *n* = 108	*p* Value
SDNN (ms)	107.5 ± 57.8	91.5 ± 44.8	0.046
Minimum HR, bpm	52.5 ± 10.9	62.1 ± 11.5	< 0.001
Maximum HR, bpm	119.5 ± 21.6	115.9 ± 17.7	0.23
Mean HR, bpm	73.7 ± 12.5	78.3 ± 10.7	0.009

## Data Availability

The raw data supporting the conclusions of this article will be made available by the authors, without undue reservation. Requests to access the datasets should be directed to cfsun1@xjtu.edu.cn; guofengwei@xjtu.edu.cn.

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
