# Peer review of "Clinical Efficacy of Catheter Ablation in the Treatment of Vasovagal Syncope"

_jcm, 2022, doi:10.3390/jcm11185371_

Round 1

Reviewer 1 Report

Dear Sir/Madam,

I had the opportunity to act as a reviewer on the recent submission by Xu et al. to the Journal of Clinical Medicine.

The authors present original research studying the efficacy and safety of catheter ablation in the treatment of vasovagal syncope. They found that high frequency stimulation mapping and anatomical ablation can effectively improve the syncope symptoms, no serious complications occurred during catheter ablation.

The manuscript is well structured; however, some issues need to be addressed:

  1. Please provide the abstract with the typical structure: background, aim, methods, results and conclusion. For instance, the aim of the study remains unclear after reading the abstract.
  2. Why was aspirin prescribed after ablation (line 160), are there any data supporting this? How about patients with known atrial fibrillation?
  3. A well-recognized problem regarding the head-up tilt test is its reproducibility: why was the change in head up tilt test from positive to negative chosen as a tool to measure effectiveness?
  4. Please provide reference for the VASIS classification (line 190) and mention it in the methods section.

Minor issues:

1-     Please rephrase: “the annular pulmonary vein catheter was punctured into the LA through the atrial septum.” (line 99). What do the authors exactly mean?

Best regards,

Author Response

我们非常感谢您的意见和建议。在修订稿中,红色字体包含包含审稿人建议的文本,并在信函中详细说明。

Reviewer 2 Report

Lingping Xu and colleagues present the clinical efficacy of catheter ablation in the treatment of vasovagal syncope (VVS). This study included 108 patients with refractory VVS who underwent catheter ablation were retrospectively enrolled. Patients were divided into two groups (i.e., high-frequency stimulation (HFS) (n=66), and anatomic landmarks 20 (n=42) targeted group) and the efficacy of the treatment was evaluated by comparing the location and probability of the intraoperative vagal reflex, the remission rate of postoperative syncope symptoms, and the rate of negative head-up tilt (HUT) results. Also the adverse events were analyzed, and safety was evaluated. During follow-up period of 8 months, both HFS mapping and anatomical ablation can effectively improve the syncope symptoms in VVS patients, and 83.7% of patients no longer experienced syncope.

Although it was a retrospective single center study, the sample size was relatively large and very informative study.

I only have a couple of minor queries:

1.     The present study is similar with the previous study which was demonstrated by Sun W et al.1) The authors should clearly demonstrate novelty and strength of the current study.  

1)     Sun W, Zheng L, Qiao Y, Shi R, Hou B, Wu L, Guo J, Zhang S, Yao Y. Catheter Ablation as a Treatment for Vasovagal Syncope: Long-Term Outcome of Endocardial Autonomic Modification of the Left Atrium. J Am Heart Assoc. 2016 Jul 8;5(7): e003471.

2.     The definition of anatomical ablation group is confusing.

The authors stated that the anatomic landmarks targeted was used when HFS failed to induce a positive reaction. And the ablation end point was the disappearance of the vagal response in all GP anatomical regions (i.e., repeated stimulation or ablation in GP no longer showed a positive response).

When HFS failed to induce GP reaction, can ablation induce the vagal response? Or can we know if a vagal response will be induced only after RF delivery?

3.     It would be more helpful in clinical practice if the authors could address the ablation parameters (i.e., RF power, RF delivery time, contact force, generator impedance drop, etc.) in both anatomical ablation group and HFS group.

Author Response

We greatly appreciate the comments and suggestions.  In the revised manuscript, the red print contains the text with suggestions of Reviewers incorporated and is detailed in the letter correspondence.

Round 2

Reviewer 1 Report

Dear Sir/Madam,

Thank you for reviewing the manuscript and addressing the mentioned issues. These were adequately answered. Therefore, the manuscript seems suitable for publishing in the present form.

Best regards